# Computer assisted detection of axonal bouton structural plasticity in *in vivo* time-lapse images

**Rohan Gala[1†], Daniel Lebrecht[2,3†], Daniela A Sahlender[4], Anne Jorstad[4], Graham Knott[4], Anthony Holtmaat[2], Armen Stepanyants[1]***

[1]Department of Physics and Center for Interdisciplinary Research on Complex Systems, Northeastern University, Boston, United States; [2]Department of Basic Neurosciences, Faculty of Medicine, University of Geneva, Geneva, Switzerland; [3]Lemanic Neuroscience Doctoral School, Switzerland; [4]Biological Electron Microscopy Facility, Centre of Electron Microscopy, École Polytechnique Fédérale de Lausanne, Lausanne, Switzerland

**Abstract** The ability to measure minute structural changes in neural circuits is essential for long-term in vivo imaging studies. Here, we propose a methodology for detection and measurement of structural changes in axonal boutons imaged with time-lapse two-photon laser scanning microscopy (2PLSM). Correlative 2PLSM and 3D electron microscopy (EM) analysis, performed in mouse barrel cortex, showed that the proposed method has low fractions of false positive/negative bouton detections (2/0 out of 18), and that 2PLSM-based bouton weights are correlated with their volumes measured in EM ($r = 0.93$). Next, the method was applied to a set of axons imaged in quick succession to characterize measurement uncertainty. The results were used to construct a statistical model in which bouton addition, elimination, and size changes are described probabilistically, rather than being treated as deterministic events. Finally, we demonstrate that the model can be used to quantify significant structural changes in boutons in long-term imaging experiments.
DOI: https://doi.org/10.7554/eLife.29315.001

*For correspondence:
a.stepanyants@neu.edu

†These authors contributed equally to this work

Competing interests: The authors declare that no competing interests exist.

## Introduction

The repertoire of synaptic connectivity within neuronal networks is immensely increased through the continuous formation and elimination of synapses (*Chklovskii et al., 2004*; *Stepanyants et al., 2002*). Indeed, in vivo imaging studies over the last 15 years have shown that synaptic structures remain dynamic throughout adulthood (*Holtmaat and Svoboda, 2009*; *Trachtenberg et al., 2002*). This structural plasticity, i.e. the appearance, disappearance, and the morphological modifications of synapses in the adult brain has been established as a fundamental underpinning of learning and experience-dependent changes in neuronal circuits (*Holtmaat and Caroni, 2016*; *Holtmaat and Svoboda, 2009*; *Stepanyants and Chklovskii, 2005*).

Synapses in the central nervous system are morphologically distinct structures, visible only in electron microscopy (EM). In light microscopy (LM) a synapse can be detected based on the presence of a swelling on the axon, referred to as bouton, or a protrusion from the dendrite, known as spine. In the cerebral cortex, the majority of excitatory synapses and a minority of inhibitory synapses occur on dendritic spines (*Gray, 1959*). Spines can easily be detected, hence most studies of structural plasticity have used manual or semi-automated tracking of these structures in time-lapse images to infer circuit changes (*Holtmaat and Svoboda, 2009*). Yet, studies relying on tracking of dendritic spines may not reveal the full extent of synaptic plasticity because synapses can also occur on dendritic shafts. On the other hand, a dendrite's presynaptic apposition can be detected as an

irregularity or swelling on the axon. Similar to dendritic spines, such axonal boutons have long since been recognized as sites of functional connections between neurons (*Van Gehuchten, 1904*). Therefore, the detection of these structures would provide a powerful means to analyze synaptic connectivity (*Markram et al., 2015*; *Meyer et al., 2010*)

Many EM studies have revealed a variety of presynaptic morphologies and the arrangements of vesicles, endoplasmic reticulum, and mitochondria therein (*Harris and Weinberg, 2012*). This is the only method capable of verifying that LM observations of axonal boutons do indeed correspond to synaptic contacts, but applying such a method every time is difficult across large volumes and impossible at more than one time point. The few real-time imaging studies of presynaptic plasticity in vivo used semi-automatic detection methods to track individual axonal boutons in LM time-lapse images (*Chen et al., 2015*; *De Paola et al., 2006*; *Grillo et al., 2013*; *Holtmaat et al., 2009*; *Johnson et al., 2016*; *Keck et al., 2011*; *Majewska et al., 2006*; *Mostany et al., 2013*; *Qiao et al., 2016*; *Stettler et al., 2006*; *Yang et al., 2016*). Visually isolated axons were segmented, and axonal boutons, as presumed synaptic contacts, were scored by virtue of their integrated fluorescence (*De Paola et al., 2006*; *Grillo et al., 2013*). Based on arbitrarily defined thresholds, boutons are usually scored in a binary fashion, i.e. present or absent. However, due to jitter in fluorescence caused by fluctuations in imaging conditions, binary scoring of boutons may lead to high false positive/negative rates.

Several computer-assisted methods aid axon segmentation (*Acciai et al., 2016*; *Parekh and Ascoli, 2013*) and bouton detection (*Song et al., 2016*) in sparsely labeled tissue, but many challenges remain. The most difficult challenges to overcome are due to the limited resolution of two-photon microscopy (mainly along the optical axis, *z*). For example, due to the high density of boutons on cortical axons, boutons often lie in close proximity to one another and can appear fused in microscopy images (*Figure 1*). In densely labeled tissue bouton detection is further confounded by the fact that axons can also appear fused to one another which locally increases the integrated fluorescence. Furthermore, spatially and temporally non-uniform expression levels and the variability in axon caliber complicate the tracking of boutons over time. Such complications lead to inconsistencies and bias in LM-based bouton detection methods. The bias will affect bouton density estimates, while the inconsistencies, combined with binary scoring of boutons, will lead to an apparent increase in the bouton turnover rate.

Here, we describe semi-automated methodology for bouton detection and tracking in images acquired by 2-photon laser scanning microscopy (2PLSM) in vivo. We validate the results of the method with correlative 3D EM of in vivo imaged axons, and by applying the detection method to images acquired under various conditions that mimic the variability in time-lapse imaging. We quantify variability in bouton detection and propose a statistical model to deal with the inherent uncertainties of this LM-based detection method.

## Results

### LM-based bouton detection and measurement of structural changes

Here, we describe and evaluate a heuristic strategy that utilizes axon traces to detect and quantify boutons in 2PLSM images taken through a cranial window in vivo. This procedure consists of the following major steps: (1) tracing axons in 3D, (2) optimization of traces, (3) generation of axon intensity profiles, (4) detection of putative boutons based on the profiles, (5) normalization of intensity profiles and calculation of bouton weights, and (6) matching putative boutons across time-lapse images.

Axons can be traced automatically or manually with various tools (*Acciai et al., 2016*; *Parekh and Ascoli, 2013*). In this study, high density of labeled axons (*Figure 1A*) precluded the possibility of automated tracing. Therefore, axons were traced manually by using NCTracer software (*Chothani et al., 2011*; *Gala et al., 2014*), and traces were optimized as described in Materials and methods. Following optimization, two intensity profiles were generated for each axon by convolving specifically designed filters with the image at all trace node positions and scaling the results to unit means. While various filters can be used to generate axon intensity profiles, in this study, we settled on the following two: (1) a modified, multi-scale Laplacian of Gaussian filter ($LoG_{xy}$) to detect putative boutons and (2) a fixed size Gaussian filter (*G*) to provide an estimate of axon intensity in the

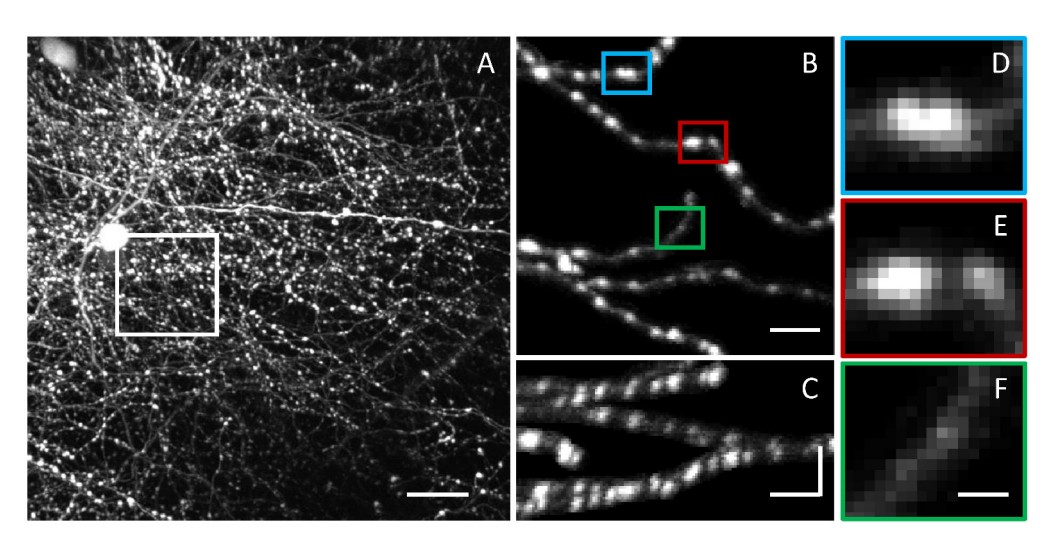

**Figure 1.** Challenges in LM-based bouton detection and measurement. (A) Maximum intensity *xy* projection of an image stack showing axons of fluorescently labeled neurons in superficial layers of mouse barrel cortex. High density of labeled axons makes it difficult to automatically detect boutons and track their structural changes over time. Scale bar is 20 μm. (B) A subset of labeled axons from the region outlined in (A). To improve visibility, image intensity beyond five voxels from the axon centerlines was set to zero. Bouton detection and bouton size measurement are confounded by large variations in fluorescence levels across axons. (C) Axons from (B) shown on the zx maximum intensity projection. Horizontal scale bars in (B) and (C) are 5 μm. Vertical scale bar in (C) is 15 μm. Lower resolution in z compared to xy is yet another challenge in bouton analyses. (D–F) Magnified views of the highlighted boutons from (B). Close proximity of boutons on an axon (D), large range of bouton sizes (E), and large range of bouton fluorescence levels (D–F), present additional obstacles to accurate bouton detection and measurement. Scale bar in (D–F) is 1.25 μm.

DOI: https://doi.org/10.7554/eLife.29315.002

---

regions devoid of boutons (see Materials and methods for details). In the following we will refer to such inter-bouton regions as axon shaft.

$$LoG_{xy}\left(x,y,z|R_{xy},R_z\right) = \frac{4e^{-(x^2+y^2)/R_{xy}^2}}{\pi R_{xy}^4}\left(1 - \frac{x^2+y^2}{R_{xy}^2}\right) \times \frac{e^{-z^2/R_z^2}}{\sqrt{\pi}R_z}$$

$$G(x,y,z|R) = \frac{e^{-(x^2+y^2+z^2)/R^2}}{(\pi)^{3/2}R^3} \tag{1}$$

*Figure 2A and B* show that distinct putative boutons can be identified as peaks in the $LoG_{xy}$ profile plotted against node positions along the trace, $I^{LoG_{xy}}(s_i)$. This is because the $LoG_{xy}$ filter is designed to sharpen boundaries between boutons by suppressing intensity in the regions immediately adjacent to boutons. In contrast, the $G$ filter yields a smoother profile, $I^G(s_i)$, which is not very useful for resolving putative boutons that are in close proximity (arrows in *Figure 2C*), but is well suited for estimating shaft intensity. For these reasons, $LoG_{xy}$ profiles were used to identify putative boutons, while $G$ profiles were used to determine intensities of axon shafts.

To automatically detect putative boutons in an $LoG_{xy}$ profile we used an algorithm that is similar to the Backward-Stepwise Subset Selection method (*Hastie et al., 2009*). Here, a varying number of foreground peaks, $N_f$, and a constant number of background peaks, $N_b$, was fitted to $I^{LoG_{xy}}(s_i)$ by minimizing the following objective function of peak positions, $\mu_j^f$ and $\mu_k^b$, amplitudes, $a_j^f$ and $a_k^b$, and widths, $\sigma_j^f$ and $\sigma_k^b$ (see Materials and methods for details):

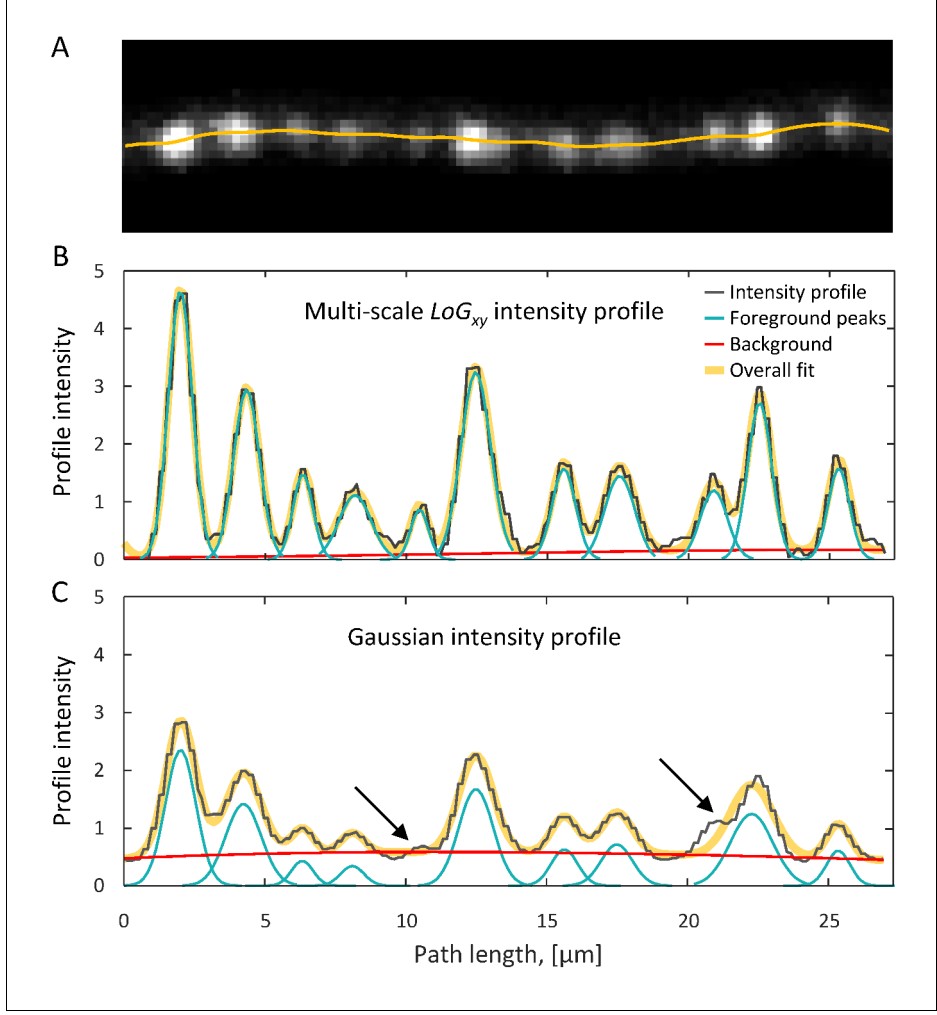

**Figure 2.** Detection of putative boutons as peaks on axon intensity profiles. (**A**) Maximum intensity *xy* projection of an axon segment showing multiple putative boutons. Yellow line is the optimized trace of this axon. (**B**) Putative boutons visible in (**A**) correspond to peaks on the *LoG$_{xy}$* intensity profile (black line). Foreground peaks (cyan lines) and local background (red line) are fitted to the intensity profile as described in the text. The overall fit (thick yellow line), which is the sum of foreground peaks and background, closely matches the intensity profile. (**C**) *G* intensity profile is obtained by sliding a fixed size Gaussian filter along the optimized trace. Small or closely positioned peaks cannot be resolved on the *G* profile (arrows). Peak amplitudes from the *LoG$_{xy}$* profile and background from the *G* profile are used to define bouton weights.

DOI: https://doi.org/10.7554/eLife.29315.003

The following figure supplements are available for figure 2:

**Figure supplement 1.** Optimization is required for trace-based bouton detection and measurement.

DOI: https://doi.org/10.7554/eLife.29315.004

**Figure supplement 2.** Trace optimization reduces variability in bouton detection and weight measurement.

DOI: https://doi.org/10.7554/eLife.29315.005

$$\min_{\substack{a_j^f, \mu_j^f, \sigma_j^f \\ a_k^b, \mu_k^b, \sigma_k^b}} \left( \sum_i \left( I^{LoG_{xy}}(s_i) - \sum_{j=1}^{N_f} a_j^f e^{-\frac{\left(s_i - \mu_j^f\right)^2}{2\left(\sigma_j^f\right)^2}} - \sum_{k=1}^{N_b} a_k^b e^{-\frac{\left(s_i - \mu_k^b\right)^2}{2\left(\sigma_k^b\right)^2}} \right)^2 \right)$$

$$a_j^f \geq 0; \quad 0.5 \mu m \leq \sigma_j^f \leq 2.0 \mu m$$

$$a_k^b \geq 0; \quad \sigma_k^b \geq 20 \mu m$$

(2)

We note that, though profile background was fit with a sum of spatially distributed peak functions, constraints imposed on their widths ($\sigma_k^b \geq 20$ μm) ensure that the fit varies slowly along the axon (red line in *Figure 2B*).

Following the detection of putative boutons, intensity of bouton $j$ was defined as the sum of the $j$-th foreground peak amplitude and background intensity at the peak location:

$$I_j^{Bouton} = a_j^f + \sum_{k=1}^{N_b} a_k^b e^{-\frac{\left(\mu_j^f - \mu_k^b\right)^2}{2\left(\sigma_k^b\right)^2}} \tag{3}$$

Shaft intensity was estimated from the $G$ profile by incorporating information about the positions of detected putative boutons, *Figure 2C*. To that end, we initialized the above described peak detection algorithm with the foreground and background peaks detected on the $LoG_{xy}$ profile, but ran the algorithm on the $G$ intensity profile. Shaft intensity for a given axon was defined as the fitted background intensity on the $G$ profile, averaged over trace nodes,

$$I_i^{Background} = \sum_{k=1}^{N_b} a_k^{b,G} e^{-\frac{\left(s_i - \mu_k^{b,G}\right)^2}{2\left(\sigma_k^{b,G}\right)^2}}$$

$$I^{Shaft} = \left\langle I_i^{Background} \right\rangle_i \tag{4}$$

In this expression, index $G$ in the superscripts of peak amplitudes, $a$, positions, $\mu$, and widths, $\sigma$, was added to emphasize that these quantities are calculated based on the $G$ intensity profile. We note that background intensity, $I_i^{Background}$, is designed to vary smoothly along the axon (red line in *Figure 2C*) and be independent of bouton density, providing a robust estimate of axon shaft intensity.

Our goal is to use intensity profiles to extract structural information related to the physical sizes of boutons. This task is hindered by the facts that axon intensity depends strongly on expression levels of fluorescent molecules (*Figure 1A*) and microscopy conditions. Therefore, to measure unbiased structural information, intensity profiles must be properly normalized. One may consider using median (or mean) profile intensity or local shaft intensity for normalization. However, these types of normalizations can lead to errors. For example, if density of boutons varies across axons, normalization with the median (or mean) may bias boutons on higher bouton density axons towards lower intensity values. Also, if density of boutons is sufficiently large, normalization with local shaft intensity can lead to variability as the latter cannot be measured reliably between closely positioned boutons.

Our heuristic normalization approach is based on the idea that by convolving the $LoG_{xy}$ or $G$ filter with the image at a trace node position $s_i$ along axon $a$ in imaging session $t$, we obtain a quantity that is proportional to three factors: $M_t$, a factor related to imaging conditions [e.g. laser power, photomultiplier tube (PMT) voltage, and cranial window quality], $\rho_{a,t}$, volume density of fluorescent molecules, and $A_{a,t}(s_i)$, a structural factor which has been linked to axon cross-section area (drawn perpendicular to the $xy$ projection of the axon centerline) convolved with the microscope point-spread function (*Song et al., 2016*) and profile filter:

$$\tilde{I}_{a,t}^{LoG_{xy}, G}(s_i) = M_t \rho_{a,t} A_{a,t}^{LoG_{xy}, G}(s_i) \tag{5}$$

In creating the $LoG_{xy}$ and $G$ profiles (*Figure 2*) we rescale $\tilde{I}_{a,t}^{LoG_{xy}}(s_i)$ and $\tilde{I}_{a,t}^{G}(s_i)$ to unit means in order to minimize effects related to imaging conditions and expression levels, thus isolating structural information:

$$I_{a,t}^{LoG_{xy}, G}(s_i) = \frac{\tilde{I}_{a,t}^{LoG_{xy}, G}(s_i)}{\left\langle \tilde{I}_{a,t}^{LoG_{xy}, G}(s_i) \right\rangle_i} = \frac{A_{a,t}^{LoG_{xy}, G}(s_i)}{\left\langle A_{a,t}^{LoG_{xy}, G}(s_i) \right\rangle_i} \tag{6}$$

It may be tempting to use putative bouton intensity, $I_j^{Bouton}$ in *Equation (3)*, which is detected based on $I_{a,t}^{LoG_{xy}}(s_i)$ as proxy for bouton size. However, this may lead to bias as the denominators in *Equation (6)* depend strongly on bouton density. To address this issue, we use axon shaft intensity detected from $G$ intensity profiles, $I^{Shaft}$ in *Equation (4)*, for normalization. The resulting quantity,

referred to as bouton weight, $w_j^{Bouton}$, conveys structural information, which is effectively independent of the above-mentioned bias,

$$w_j^{Bouton} = \frac{I_j^{Bouton}}{I^{Shaft}} \tag{7}$$

## Validation of LM-based bouton detection methodology with EM

Correlative light and electron microscopy (CLEM) was used to validate the described bouton detection procedure (*Figure 3* and *Figure 3—video 1*). Four axon segments were selected for this analysis (*Figure 3A*). The axon segments were imaged in vivo with 2PLSM, the brain tissue was fixed shortly after, and subsequently imaged with EM (see Materials and methods for details). Putative boutons in the 2PLSM stack of images were detected and quantified as described above (*Figure 3B*). Axons in EM images were reconstructed in 3D (*Figure 3C*) and rendered in Blender software for further analysis.

Putative boutons detected in 2PLSM images could be unambiguously matched with varicosities identified in EM (*Figure 3—video 1*, asterisks in *Figure 4A–D*, and *Table 1*). We used positions of the centers of matched boutons to register LM traces and EM centerlines with an optimal linear transformation (*Hastie et al., 2009*). Average distance between the registered traces (*Gala et al., 2014*) was small, 0.34 ± 0.17 µm (mean ± s.d.), confirming that the same set of axons was reconstructed in LM and EM. A small discrepancy between registered traces may be attributed to non-linear distortion of tissue in the EM experiment and the relatively low *z*-resolution of LM. Overall, 16 boutons *en passant* and one bouton *terminaux* could be identified in EM. The bouton *terminaux* was detected with our method, but was excluded from further analysis because it is not located on the axon centerline. With the omission of this bouton, the LM-based procedure detected 18 putative boutons with 0 false negatives and two false positives, both of which were small (1.1 and 2.2 in weight, #18 and #19 in *Table 1*).

We tested the idea that 2PLSM intensity is correlated to the area of axon cross-section drawn perpendicular to the *xy* projection of axon centerline (*Song et al., 2016*). We refer to such cross-sections as *z* cross-sections (third row in *Figure 4A–D*) to distinguish them from normal cross-sections, which are drawn perpendicular to the axon centerline. For this, we calculated the normalized axon intensity profile, $I_{a,t}^{LoG_{xy}}(s_i)/I^{Shaft}$, and compared it to EM *z* cross-section areas including and excluding mitochondrial *z* cross-section areas. Both *z* cross-section areas appear to be well correlated with normalized intensity profiles. However, it is clear from visual inspection that the normalized intensity profiles do not resolve small changes in axon cross-section, which is the result of limited resolution of 2PLSM.

To examine the extent to which LM-based measurements provide information about bouton size, we plotted bouton weight against bouton volumes including (*Figure 4F*) and excluding (*Figure 4G*) mitochondrial volume. Normal axon cross-sections were used to identify bouton boundaries and calculate bouton volume as described in Materials and methods (fourth row in *Figure 4A–D* and *Figure 4E*). Bouton #7 (red box in *Figure 4B*) was excluded from these analyses, because it is bounded on one side by the terminal bouton branch which biases weight measurement. The results show high degree of correlation in both cases (Pearson's $r = 0.93$), supporting the idea that LM-based measurements can be used to quantify volumes of even very small varicosities (#3, 0.093 µm³). The results also provide support for the choice of filters, *Equation (1)*, and the normalization procedure, *Equation (7)*, showing that the proposed method could overcome axon-specific differences in the expression levels and bouton density, capturing meaningful fine-scale structural information.

It is important to emphasize that we did not attempt to find the best procedure for fitting the EM data. It is possible that some other combinations of filter types, normalizations, and parameters would lead to marginally better correlations. However, due to the small sample size, this would likely be a case of over-fitting. The described procedure was designed based on theoretical considerations, and the parameters were chosen based on the observed range of bouton sizes (see Materials and methods). Nonetheless, we explored alternative filtering and normalization strategies. *Figure 4—figure supplement 1* shows profiles derived from raw voxel intensities, and profiles obtained with mean, Gaussian, and median filters of different but fixed sizes. Each of these profiles was normalized by its median value. The figure makes it clear that some amount of filtering benefits bouton

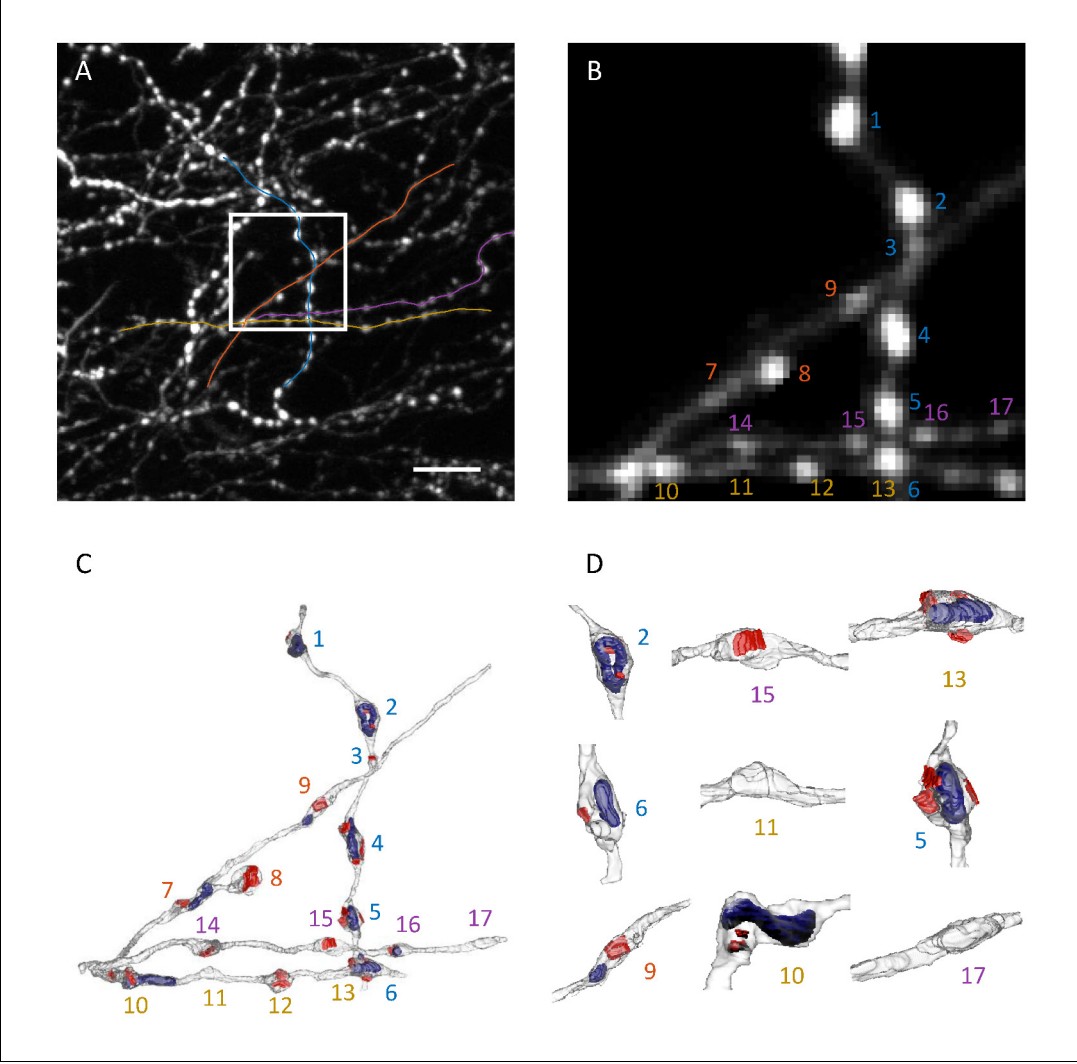

**Figure 3.** Correlative light and electron microscopy. (**A**) Maximum intensity *xy* projection of an image stack used for CLEM analysis. White box demarcates the region imaged with EM. Colored lines are traces of four axon segments chosen for EM reconstruction. Scale bar is 10 μm. (**B**) Region outlined in (**A**) is shown at 4x magnification with background removed. (**C**) 3D EM reconstruction of the four axon segments shown in (**B**). Red areas mark PSDs, and blue volumes outline mitochondria. Most varicosities identified in EM are clearly visible in 2PLSM images (**B**). (**D**) Higher magnifications and different orientations of a subset of reconstructed varicosities shows that structural swellings on axons may or may not be associated with PSDs and/or contain mitochondria. Numbers in (**B–D**) enumerate distinct varicosities identified in EM.

DOI: https://doi.org/10.7554/eLife.29315.006

The following video is available for figure 3:

**Figure 3—video 1.** Illustration of correlative light and electron microscopy analysis.

DOI: https://doi.org/10.7554/eLife.29315.007

detection, as jagged unfiltered profiles can lead to false positives. Similarly, relatively large filters of fixed size (e.g. $5 \times 5 \times 5$ and $R = 2$) are not well suited for bouton detection as they often suppress intensities of small boutons and merge closely positioned boutons. When the filter size is carefully tuned ($3 \times 3 \times 3$ and $R = 1$), resulting profiles can look similar to those of multi-scale $LoG_{xy}$, although higher baselines in these profiles can make peak detection more challenging. Furthermore, we examined the combined effect of filtering and normalization on bouton measurement by comparing the amplitudes of profile peaks to corresponding bouton volumes. Multi-scale $LoG_{xy}$ profile

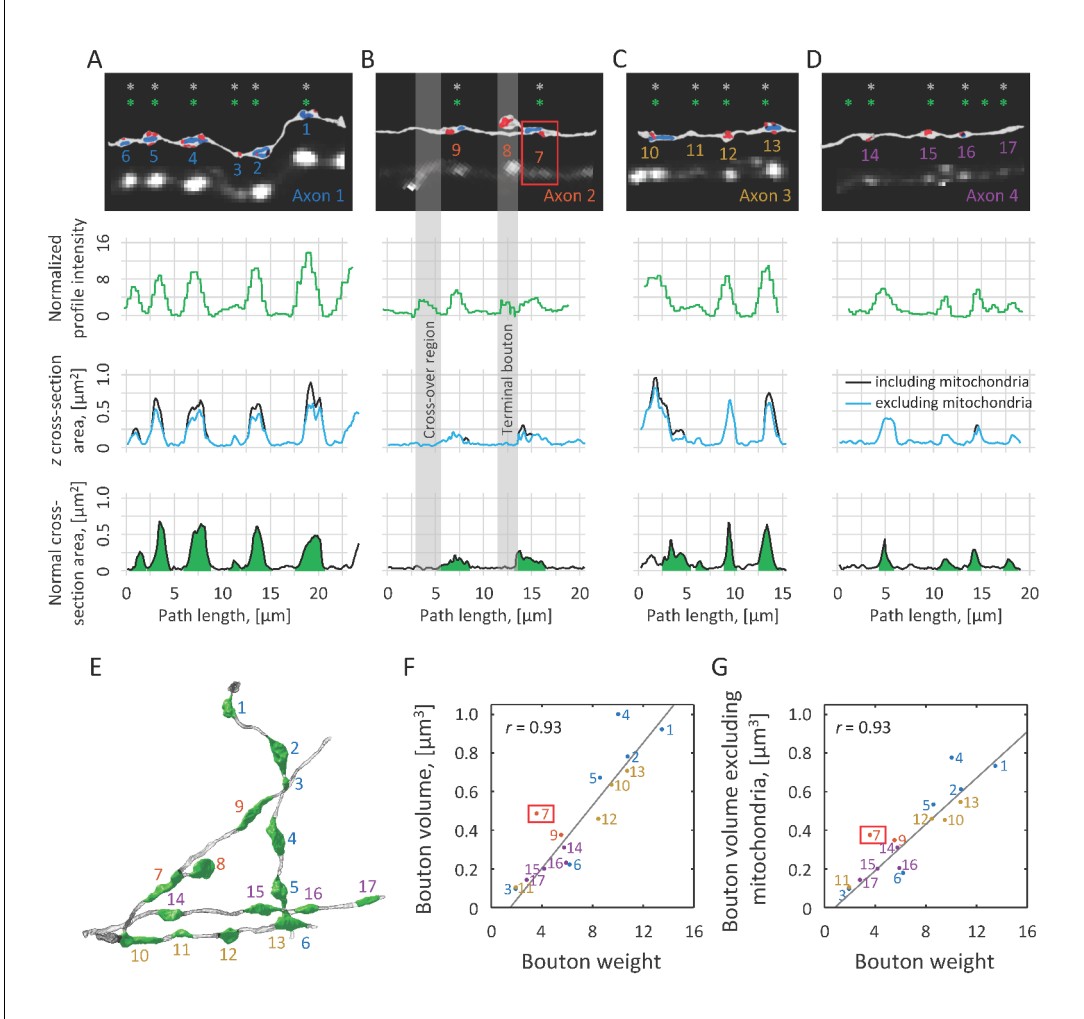

**Figure 4.** Normalized intensity profile is correlated with axon cross-section area, while bouton weight is indicative of bouton volume. (**A–D**) Top row: EM reconstructions of axons are shown next to the corresponding 2PLSM maximum intensity projections. Varicosities identified in EM are marked with grey asterisks, and putative boutons automatically detected based on the intensity profiles are marked with green asterisks. Second row: Normalized intensity profiles of the same axons plotted as functions of position along axon centerlines obtained in EM. Third row: $z$ cross-section areas of axons, including (black) and excluding (blue) mitochondria, are well correlated with intensity profiles. These cross-sections are drawn perpendicular to the 2PLSM $xy$ projections of the axon centerlines. Fourth row: Normal cross-section areas showing the extents of boutons (green) based on criteria described in Materials and methods. (**E**) Demarcated boutons shown in 3D (green). Bouton weights are highly correlated with their EM-based volumes that include (**F**) or exclude (**G**) mitochondrial volumes. Grey regions in (**B**) highlight two spurious peaks in the normalized intensity profile, one resulting from a close apposition of two axons and another caused by the presence of a terminal bouton. Such regions were annotated in 2PLSM images and were excluded from all analyses. In addition, bouton #7 (red box in B, F, and G) was excluded from the correlation analysis because it is directly adjacent to a branch point, which biases weight calculation.
DOI: https://doi.org/10.7554/eLife.29315.008

The following figure supplement is available for figure 4:

**Figure supplement 1.** Filter type, filter size, and profile normalization can affect bouton detection and measurement.
DOI: https://doi.org/10.7554/eLife.29315.009

normalized with shaft intensity as described above leads to the highest correlation compared to the best considered alternatives (*Figure 4—figure supplement 1E*).

## Effects of imaging conditions on bouton analysis
Next, we sought to evaluate the effects of various imaging conditions on bouton detection and measurement. To that end, a set of fluorescently labeled axons was imaged seven times in a span of 80

**Table 1.** Comparison of LM-based and EM measurements.

Bouton IDs match those in **Figures 3** and **4**. The probability that a putative bouton belongs to the category of LM boutons, $P(bouton|w)$ was calculated according to **Equation (9)** with $w_{threshold}$ = 2.0. Bouton #8 (grey) was excluded from the analyses as it is a terminal bouton. Hyphens indicate that boutons, mitochondria, or PSDs were not detected in EM.

| | Bouton ID | 2PLSM measurements | | EM measurements | | |
| --- | --- | --- | --- | --- | --- | --- |
| | | Putative bouton weight, $w$ | $P(bouton \mid w)$ | Bouton volume [$\mu m^3$] | Mitochondria volume [$\mu m^3$] | PSD surface area [$\mu m^2$] |
| Axon 1 | 1 | 13.5 | 1.00 | 0.919 | 0.189 | 0.666 |
| | 2 | 10.8 | 1.00 | 0.779 | 0.170 | 0.595 |
| | 3 | 1.98 | 0.48 | 0.093 | - | 0.371 |
| | 4 | 10.1 | 1.00 | 0.998 | 0.225 | 1.92 |
| | 5 | 8.65 | 1.00 | 0.669 | 0.139 | 1.83 |
| | 6 | 6.25 | 1.00 | 0.219 | 0.043 | 0.138 |
| Axon 2 | 7 | 3.63 | 0.93 | 0.483 | 0.111 | 0.612 |
| | 8 | N/A | N/A | 0.574 | - | 2.75 |
| | 9 | 5.57 | 1.00 | 0.372 | 0.027 | 1.28 |
| Axon 3 | 10 | 9.54 | 1.00 | 0.632 | 0.182 | 0.645 |
| | 11 | 1.99 | 0.49 | 0.102 | - | - |
| | 12 | 8.51 | 1.00 | 0.456 | - | 1.23 |
| | 13 | 10.8 | 1.00 | 0.704 | 0.161 | 1.49 |
| Axon 4 | 14 | 5.80 | 1.00 | 0.308 | - | 0.867 |
| | 15 | 4.23 | 1.00 | 0.198 | - | 0.686 |
| | 16 | 5.96 | 1.00 | 0.229 | 0.027 | 0.402 |
| | 17 | 2.85 | 0.93 | 0.140 | - | - |
| | 18 | 1.14 | 0.01 | - | - | - |
| | 19 | 2.19 | 0.65 | - | - | - |

DOI: https://doi.org/10.7554/eLife.29315.010

min with different laser power (LP) and PMT voltage (see inset in **Figure 5A**). In addition, in condition E, a thin layer of agarose was applied to the cranial window to mimic deterioration of window quality which often accompanies long-term imaging experiments. In the following, we assume that there is negligible structural plasticity of boutons throughout the duration of this short-term imaging experiment, and therefore differences in bouton measurements can be attributed to the effects of imaging conditions and measurement uncertainties. The inset in **Figure 5B** shows the maximum intensity projections of an axon segment imaged in all seven conditions. This inset illustrates that increasing (decreasing) LP and/or PMT voltage results in an overall increase (decrease) in intensity, and therefore proper normalization procedure must be used to minimize the effects of such changes on bouton measurements.

Putative boutons on 16 axon segments were detected in all conditions independently (400 putative boutons per condition on average). Custom software was used to match the same putative boutons across conditions. Putative boutons detected in condition A were chosen to be the gold standard, and precision/recall in bouton detection in the remaining conditions B-G were evaluated as functions of bouton weight (**Figure 5A and B**). The results show that for 95% of putative boutons of weight $w > 2.0$ both precision and recall equal one in all conditions. This number of unambiguously detected putative boutons goes up to 99% for $w > 2.5$ and 100% for $w > 3.1$. For reference, $w = 2.0$ corresponds to the weights of the two smallest varicosities identified in EM (#3 and #11 in **Table 1**), and the weight of the next smallest bouton (#17) is $w = 2.8$. Therefore, all but very small boutons can be detected with our method with high confidence.

To examine the effects of imaging conditions on bouton weight, we tested a subset of 290 putative boutons that were detected and matched across all seven experiments. **Figure 5C** shows bouton weights in conditions B-G plotted against the corresponding weights in condition A. Although

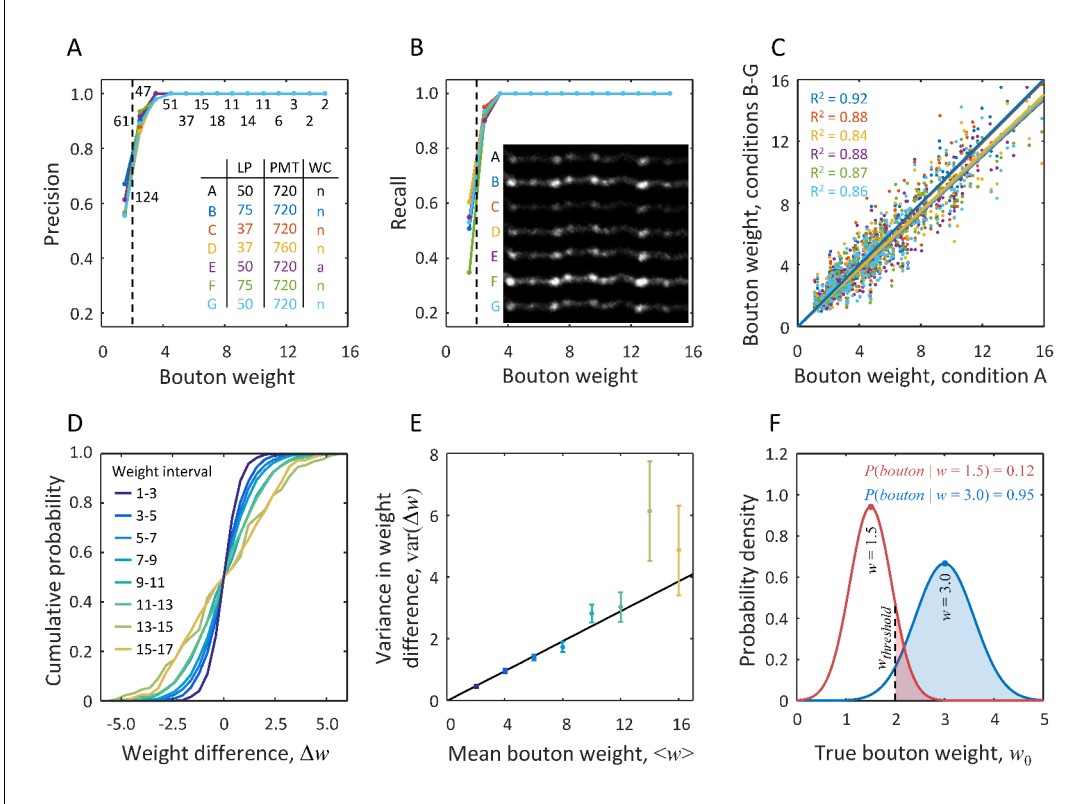

**Figure 5.** Probabilistic definition of an LM bouton based on measurement uncertainty derived from short-term imaging experiments. The same set of axons was imaged 7 times within 80 min with various microscope settings and cranial window conditions (inset in A). Putative boutons detected based on the first imaging session (condition A) were chosen to be the gold standard. Precision (A) and recall (B) in bouton detection were measured under the remaining conditions, B-G. Both precision and recall increase with bouton weight. While for very small boutons ($w < 2.0$, dashed line) detection is unreliable, agreement with the gold standard is achieved across all imaging conditions in 95% of boutons with weights greater than 2.0. Numbers of boutons in the gold standard are indicated next to the data points in (A). Inset in (B) shows an example of one axon segment imaged in conditions A-G. (C) Bouton weights under different imaging conditions are plotted against the gold standard weight. Best fit lines show no significant bias for conditions B and C, however small, but significant reduction in mean bouton weight was observed in the remaining four conditions (all $p < 0.03$, two-sample $t$-test). Abbreviations used in the inset of A: LP is laser power in mW, PMT denotes photomultiplier tube voltage in Volts, and WC is cranial window condition, where 'n' stands for normal and 'a' indicates presence of a thin layer of agarose. Color code used in (A–C) is defined by the inset table in (A). (D) CDFs for differences in bouton weights across imaging conditions. Data from all conditions were pooled. Different lines show CDFs for various intervals of mean bouton weight. (E) Variance in bouton weight difference increases linearly with mean bouton weight ($\chi^2$ linear regression with $\mathrm{var}(\Delta w) = \alpha\langle w\rangle$, $p = 0.33$, $\alpha = 0.24 \pm 0.01$, mean ± s.d.). Error-bars indicate standard deviations obtained with bootstrap sampling with replacement. (F) Red line shows the distribution of true bouton weight for a putative bouton of measured weight $w = 1.5$. Area under the curve to the right of $w_{threshold} = 2.0$ gives $P(bouton|w) = 0.12$. Large putative boutons (e.g. blue curve, $w = 3.0$) have high probability of being LM boutons.

DOI: https://doi.org/10.7554/eLife.29315.011

axon intensities in conditions B and C are drastically different from A (see inset in *Figure 5B*), the normalization procedure could correct for these differences (blue and red lines in *Figure 5C*). However, small ($\approx 7\%$) but significant bias was present in conditions D-G, which may have been caused by bleaching due to prolonged imaging. *Figure 5C* also reveals a considerable amount of variability in weight measurements. This variability, reflected in the $R^2$ coefficients, was similar across all comparisons, including the imaging experiments performed under the same conditions (e.g. A and G, cyan points). Therefore, bouton weight measurements are accompanied with uncertainty which is inherent to the LM-based methodology. This uncertainty cannot be eliminated entirely, and hence, it must be explicitly incorporated into models that derive biological information from bouton measurements.

## Statistical framework for the analysis of structural plasticity of boutons

To model the variability observed in *Figure 5C*, we examined bouton weight differences for all pairs of imaging conditions, $\Delta w = w_1 - w_2$. Because such changes clearly depend on bouton size, we looked at the statistics of $\Delta w$ in various intervals of average bouton weight, $\langle w \rangle = (w_1 + w_2)/2$. *Figure 5D* shows the cumulative distribution functions (CDFs) of bouton weight differences in eight intervals of $\langle w \rangle$. These distributions are not significantly different from Gaussian distributions, and their variances are roughly proportional to the mean bouton weights, $\mathrm{var}(\Delta w) = \alpha \langle w \rangle$, *Figure 5E*. Therefore, all CDFs shown in *Figure 5D* could be standardized by rescaling the weight differences as $\Delta w / \sqrt{\alpha \langle w \rangle}$, leading to distributions that are statistically indistinguishable from the Standard Normal CDF (all p > 0.05, one-sample KS test).

Based on this result, we propose a statistical model in which measured bouton weight, $w$, is the sum of the true weight, $w_0$, and a noise term, $\delta$. The latter is randomly drawn from a Gaussian distribution with variance proportional to the measured weight, $\mathrm{var}(\delta) = \frac{1}{2}\alpha w$:

$$w = w_0 + \delta; \quad P(\delta) = \frac{e^{-\delta^2/\alpha w}}{\sqrt{\pi \alpha w}} \tag{8}$$

*Equation (8)* quantifies uncertainties in bouton weight measurements, making it possible to define bouton presence probabilistically. To that end, we impose a threshold on true bouton weight, $w_{threshold}$, and refer to putative boutons with $w_0 > w_{threshold}$, as LM (light microscopy) boutons. In the following we set $w_{threshold} = 2.0$, which is motivated by several considerations. First, this value equals twice the average normalized axon shaft intensity. Therefore, by using $w_{threshold} = 2.0$ we are only including peaks that are substantially larger than the axon shaft intensity. Second, $w = 2.0$ corresponds to the weights of the two smallest varicosities detected in EM (#3 and #11 in *Table 1*). Finally, detection of putative boutons becomes unreliable for $w < 2.0$ (*Figure 5A and B*). We note that as an alternative to choosing a single threshold value to define LM boutons, one could vary the threshold in a certain range (e.g. 1–3) and report results as functions of this parameter.

The probability that a putative bouton of measured weight $w$ belongs to the category of LM boutons, $P(bouton|w)$, can be calculated based on the noise model of *Equation (8)*:

$$P(bouton|w) = \frac{1}{2}\left(1 + erf\left(\frac{w - w_{threshold}}{\sqrt{\alpha w}}\right)\right) \tag{9}$$

In this expression, *erf* denotes the error function. Shaded regions in *Figure 5F* illustrate these probabilities for two putative boutons of measured weights $w = 1.5$ and $w = 3.0$. Note that even when $w$ is less than $w_{threshold}$, there is a non-zero probability that the detected peak is an LM bouton. For large $w$ (e.g. greater than 3), this probability approaches unity, and the LM bouton definition becomes virtually deterministic.

LM bouton definition can be used to calculate the probabilities of bouton addition, elimination, potentiation, and depression based on the measured weights in two imaging sessions (initial and final), $w_i$ and $w_f$:

$$P(added|w_i \to w_f) = (1 - P(bouton|w_i)) \times P(bouton|w_f)$$

$$P(eliminated|w_i \to w_f) = P(bouton|w_i) \times (1 - P(bouton|w_f))$$

$$P(potentiated|w_i \to w_f) = P(bouton|w_i) \times P(bouton|w_f) \times \frac{1}{2}\left(1 + erf\left(\frac{w_f - w_i}{\sqrt{\alpha(w_i + w_f)}}\right)\right) \tag{10}$$

$$P(depressed|w_i \to w_f) = P(bouton|w_i) \times P(bouton|w_f) \times \frac{1}{2}\left(1 + erf\left(\frac{w_i - w_f}{\sqrt{\alpha(w_i + w_f)}}\right)\right)$$

We would like to clarify that LM boutons may or may not correspond to varicosities seen in EM (*Table 1*), and the latter may not always be associated with postsynaptic densities (PSDs) and, thus, functional synapses (*Shepherd and Harris, 1998*). Therefore, the relationship between an LM bouton and a synapse is not deterministic and is likely to contain false-positives and false-negatives. However, the number of such errors is expected to decrease with increasing $w$. For example, *Table 1*

shows that all LM boutons of $w > 2.2$ correspond to EM varicosities, and all LM boutons of $w > 2.9$ are varicosities associated with PSDs.

## Bouton changes can be resolved despite the uncertainty in measurements

Next, we set out to determine if the above described bouton detection procedure can be used to identify significant structural changes in long-term, in vivo imaging experiments. To that end, axons of GFP labeled neurons were imaged in superficial layers of barrel cortex in five mice. Imaging was performed at 4 day intervals over a 24 day period (seven imaging sessions). On average, 968 bouton sites were detected on 20 axon segments in each animal and imaging session. These sites were tracked over the duration of the experiment (*Figure 6—figure supplement 1*) to quantify bouton addition, elimination, and weight changes as compared to the initial state of the circuit. The short-term imaging experiment of *Figure 5* was used as control. Here, conditions B-G were compared to condition A, and the results were pooled.

*Figure 6A* shows the fraction of added LM boutons over time, as compared to the first imaging session. Specifically, we only consider significant bouton addition events, i.e. those for which the probability of addition according to *Equation (10)* is greater than 0.95. Analogous plots for the fractions of significant bouton eliminations and significant bouton weight changes (potentiation and depression combined) are shown in *Figure 6B and C* respectively. As was expected, the fractions of significant changes in the short-term imaging experiment (red points in *Figure 6*) are at the chance levels (dashed red lines). The latter was obtained with a bootstrap procedure in which the weights of individual boutons were independently shuffled across conditions. In contrast, in the long-term imaging experiment, the fractions of significant bouton additions, eliminations, and weight changes are significantly larger than chance already by the second imaging session (after 4 days), and these fractions continue to increase with time, consistent with the idea of gradual modification of the circuit.

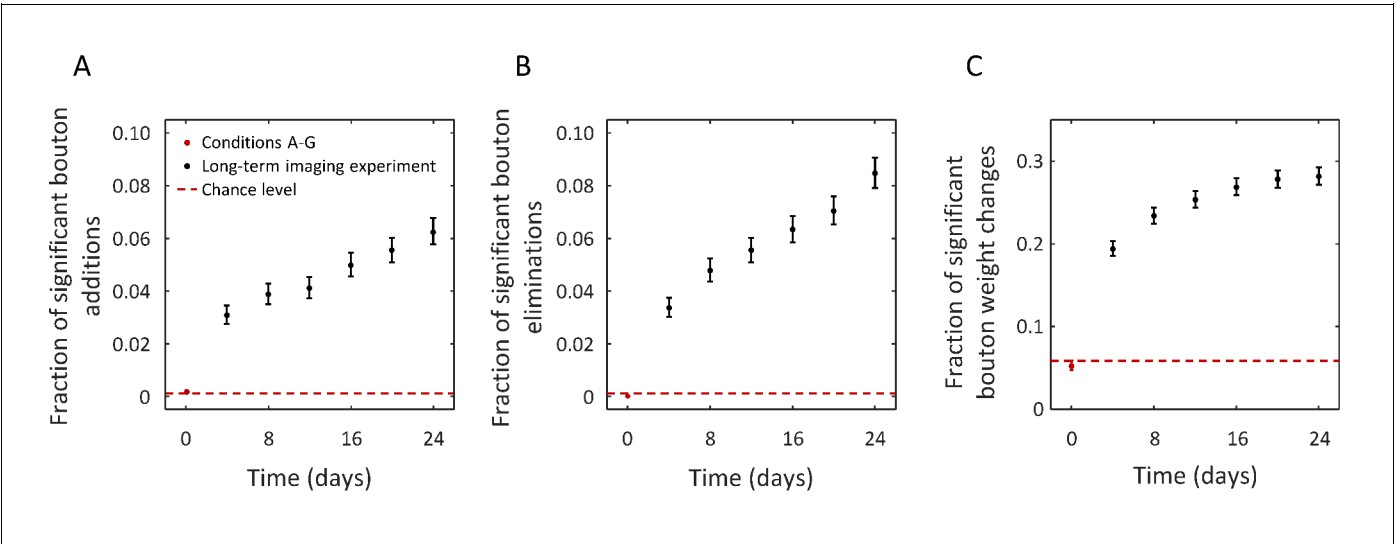

**Figure 6.** Structural change in boutons can be resolved in long-term in vivo imaging experiments despite measurement noise. (**A**) Fraction of boutons that are absent on day 0 and are present at a later time with joint probability of 0.95 or greater (significant bouton addition). Red point (error-bars are too small to be visible) shows this fraction in images acquired within 80 min (conditions A-G). Black points show the results for a long-term imaging experiment. Statistically significant fraction of added boutons is detected after 4 days (interval between imaging sessions), and this fraction grows with time, consistent with the idea of gradual modification of the initial circuit. Similar trends were observed for the fractions of significant bouton eliminations (**B**) and significant bouton weight changes. Dashed red lines in (**A–C**) indicate baseline circuit changes expected from the statistical model. Error-bars indicate standard deviations based on Poisson statistics.

DOI: https://doi.org/10.7554/eLife.29315.012

The following figure supplement is available for figure 6:

**Figure supplement 1.** Matching boutons in time-lapse images.

DOI: https://doi.org/10.7554/eLife.29315.013

These results illustrate that the described methodology can be used to quantify circuit changes, despite the numerous challenges associated with long-term, in vivo imaging experiments.

## Discussion

Detection of structural changes in boutons is hindered by various technical challenges and fundamental limitations of light microscopy. Unfortunately, any uncertainty that enters the analysis of long-term in vivo imaging data manifests itself as spurious structural plasticity. Therefore, it is important to account for all sources of errors, minimize their effect, and incorporate the residual uncertainty into the interpretation of results. Below, we describe various sources of errors affecting bouton measurements.

First, fluorescence based measurements provide only indirect evidence of bouton size. Therefore, such measurements need to be validated by showing that they are informative of bouton structures. In this study, we used CLEM to show that bouton weight, defined based on 2PLSM data, is well correlated with bouton volume (*Figure 4F and G*). Second, variability in expression levels of fluorescent proteins across axons and within axons over time makes it difficult to compare boutons on different axons and to identify true structural changes. Here, these problems were mitigated by a specifically designed normalization procedure (*Figure 2*), which was verified with CLEM (*Figure 4*) and was tested in the short-term imaging experiment (*Figure 5A–C*). Third, because the resolution of 2PLSM ($\approx$ 0.3 μm in *xy*) is not much smaller than bouton size ($\approx$ 1 μm for small boutons in *Figure 3*), there are inherent uncertainties in detection of small boutons, differentiation of closely positioned boutons, and measurement of bouton size changes. These uncertainties of the imaging method cannot be eliminated, and, therefore, they must be modeled to ensure that bouton measurements are interpreted correctly (*Figure 5D–F*).

Many other technical issues can add to the uncertainty and introduce bias in bouton measurements. For example, deterioration of cranial window quality with time, deformation of brain tissue, small changes in brain orientation relative to the microscope, and slight movement of boutons along axons can lead to the perception of increased plasticity. In contrast, limited temporal resolution of long-term imaging experiments, typically few days between imaging sessions, can lead to an underestimate of plasticity, as changes that occur between imaging sessions remain unaccounted. Finally, uncertainty can arise from the computational method used to detect boutons. *Figure 2—figure supplement 2D* shows that computational uncertainty of the method described in this study is small and can be ignored considering contributions from other sources (*Figure 5C*).

It is important to emphasize that not every putative bouton detected in LM will correspond to a varicosity if imaged with EM. This is particularly so for very small putative boutons ($w < 2.0$, bouton volume $< 0.1$ μm$^3$) which can result from noise in fluorescence or inhomogeneity in labeling. In the absence of a reliable synaptic marker, the only reasonable way to eliminate such functionally irrelevant putative boutons is by imposing a threshold on bouton weight. In this study, we refer to putative boutons whose true weight exceeds the threshold as LM boutons and suggest setting this threshold at 2.0 based on CLEM and short-term imaging experiments. In practice, one may vary the threshold in a small range to ensure that specific conclusions are robust to the choice of this parameter.

Using a threshold to define an LM bouton as an all-or-none entity may impose a bias since the true bouton weight is unknown due to measurement uncertainty. The problem is exacerbated by the unimodal shape of bouton weight distributions (see *Figure 2—figure supplement 2B*), because of which a large fraction of weights will lie within the range of uncertainty around the threshold, regardless of its value. Such ambiguous boutons can either be discarded, which may bias biological interpretation, or, otherwise, they must be treated probabilistically. For example, probabilistic description is essential for calculation of the expected number of LM boutons. Simply counting the number of above threshold weights would result in an underestimate as bouton weight distribution is monotonically decreasing. Similar considerations apply to calculations of expected numbers of added, eliminated, potentiated, and depressed boutons. As most bouton structures are stable and do not change substantially over days-to-weeks, their weight changes often lie within the range of measurement uncertainty and must also be described probabilistically. Furthermore, probabilistic description is necessary for calculation of error-bars and for establishing statistical significance of results. To illustrate the feasibility of this approach, we applied it to a dataset of 4840 bouton sites

tracked over 24 days. *Figure 6* shows that statistically significant changes in LM boutons can be detected in long-term imaging experiments. It remains to be seen if the detected changes are informative of circuit alterations that subserve the many functions of the brain.

## Materials and methods

### In vivo imaging

All experiments were performed according to the guidelines of the Swiss Federal Act on Animal Protection and Swiss Animal Protection Ordinance. The ethics committee of the University of Geneva and the Cantonal Veterinary Office of Geneva, Switzerland (approval code GE/61/17) approved all experiments.

For development of the detection methodology we used data from mice that had been repeatedly imaged as part of a larger optical micro-stimulation and behavioral study (not yet published). Only the methodology that is relevant for the present study will be mentioned. Adeno-associated viral (AAV) vectors encoding floxed GFP (AAV2/9.CAG.flex.eGFP.WPRE.bGH2 [Upen]) were co-injected with an AAV vector encoding Cre (AAV2/9.hSynapsin.hGHintron.GFP-Cre.WPRE.SV40 [Upen]) at a ratio of $0.15 \times 10^9$: 1 (genome copies: genome copies). This produced an average of about 150 GFP-expressing cells per animal, mostly in L2/3 and L5 of the barrel cortex. We exclusively imaged the processes of these neurons in the upper layers of the cortex. Hence, the source of the fluorescent signal in the green channel is dominated by cytosolic GFP.

Following the AAV injection, we implanted a 3 mm diameter glass window over the barrel cortex as previously described (*Holtmaat et al., 2009*). Imaging experiments were started 6 weeks after the virus injection. Imaging was performed under anaesthesia (0.5–1.5% isoflurane). A custom-built alignment system was used to ensure identical positioning of the mouse's cranial window at different time points, and to avoid rotations relative to the axial dimension of the objective. The system was based on aligning a beam reflected by the cranial window through two apertures, which uniquely determines a line in three spatial dimensions. The apertures and the laser position were constant, and the only free variables were the cranial window position and rotation (i.e., the *x*, *y*, and *z*-positions, as well as the row and pitch needed to be identical between sessions to satisfy the passage of light through the two apertures). We used a custom built 2PLSM (Janelia Research Campus, model Non-MIMMS in vivo microscope) to acquire in vivo anatomical images of the anesthetized animals through the cranial window. We used the ScanImage software (Janelia Research Campus, Vidrio Technologies) (*Pologruto et al., 2003*) to record the 2PLSM images, and additional custom software to align anatomical structures present on the central image stack by using a red-green overlay between current and past images.

For imaging we used a 20x water immersion objective (NA 0.95, XLUMPlanFI, Olympus, Japan). Images were acquired at a voxel volume of $\approx 0.13 \times 0.26 \times 0.8$ μm³ ($x \times y \times z$). We binned the data in the *x* dimension to generate isometric voxels in *x* and *y* for all analyses. The voxel dwell time was 0.8 μs for the full resolution images (and thus twice as high after 2x binning of the *x* dimension). Each ROI comprised an image stack of $\approx 270 \times 270 \times 250$ μm³. As the imaging light source we used a Ti:sapphire femtosecond pulsed laser (Chameleon ultra II, Coherent) that was lasing at 1010 nm. Emitted fluorescence light was split into two channels using a dichroic mirror (Semrock, FF735-Di01−35.5 × 49.0) and two bandpass filters (red channel, Semrock FF01-607/70; green channel, Semrock FF01-530/70). Each channel was equipped with a PMT (red channel, Hamamatsu R3896; green channel, Hamamatsu H10770PA-40SEL). Images were analyzed only in the green channel.

### Correlative 3D EM

Electron microscopy was carried out on the region outlined in *Figure 3A* using a previously described method (*Maco et al., 2014*; *Maco et al., 2013*). An image of the blood vessel pattern below the cranial window was taken immediately after the last 2PLSM imaging session. The anesthetized animal was then chemically fixed by perfusing, via the heart, 10 ml of isotonic PBS immediately followed by 200 ml of 2.5% glutaraldehyde and 2% paraformaldehyde in phosphate buffer (0.1 M, pH 7.4). Two hours later, the brain was removed and 60-μm-vibratome sections cut from the imaged region, tangential to the cortical surface. The region containing the imaged axons was located in these sections by matching the pattern of blood vessels seen in the first 2–3 sections with the image

of the brain's surface taken prior to the perfusion. The 2-photon laser was then used to burn lines into the fixed section around the region of interest. The laser was tuned to $\lambda$ = 850 nm with a power of ~300 mW at the back focal plane of the objective. A rectangle of ~25 μm x 25 μm was burnt around the ROI using the line scan mode (2 ms/line). This shape acts as a fiducial mark that can be seen after the tissue has been stained, and resin embedded, ready for election microscopy. This section was then washed in sodium cacodylate buffer (0.1 M, pH 7.4) 5 × 3 min, postfixed and stained in reduced osmium (1.5% potassium ferrocyanide with 1% osmium tetroxide in 0.1 M sodium cacodylate buffer) for 40 min, followed by another 40 min incubation with 1% osmium tetroxide in the same buffer, and finally for 40 min with 1% aqueous uranyl acetate. Next, the section was dehydrated in a series of increasing concentrations of alcohol, then infiltrated in 100% Durcupan resin, and flat embedded between two glass slides and placed in a 60°C oven for 24 hr.

A small piece of the section (2 mm x 2 mm), containing the laser marked region, was trimmed from the rest of the section and glued onto a slab of blank resin, keeping the region of interest closest to the surface. This block was then trimmed in the ultramicrotome, using glass knives, so that the laser marks were within 5 μm of its surface, and less than 2 μm from one edge. The block was then mounted on a 45° aluminum stub, using conductive carbon paint, so that this edge was uppermost. It was then sputter coated with a 30 nm thick layer of gold. The sample was then placed in the focused ion beam scanning electron microscope chamber (FIBSEM; Zeiss NVision 40, Zeiss SMT, Germany) and orientated so that the ion beam could mill parallel to the laser marks, and therefore parallel to the 2PLSM plane of focus. The sample was imaged with an electron beam of 1.7 kV and a probe current of 1.1 nA using the back-scattered electron detector (ESB). Images with a pixel size of 6 nm were collected with a dwell time of 12 μs per pixel. The ion beam removed 12 nm of resin after each image, using a current of 700 pA, with an energy of 30 kV.

The serial electron micrographs were aligned in FIJI (http://fiji.sc; *Schindelin et al., 2012*) and the structures of interest segmented in the TrakEM2 part of the software (*Cardona et al., 2012*). Models were then exported to the Blender software (Blender Foundation, Amsterdam; http://www.blender.org), and the NeuroMorph toolset (http://neuromorph.epfl.ch; *Jorstad et al., 2015*) was used to measure axon cross-section areas, bouton and mitochondrial volumes, and PSD surface areas. The beginning and end of each bouton were defined, using the centerline processing tool, as the points along the axon where its normal cross-section area changed by more than 30% within a distance of 0.2 μm. The boutons were then delineated and their volumes calculated in the measurement tool (fourth row in *Figure 4A–D* and *Figure 4E*).

## Trace optimization

*Figure 2—figure supplement 1A* shows three maximum intensity projection views of an axon segment, which was traced independently by five users. Inter-user trace variability, which is usually more pronounced along the z-dimension (optical axis), can hinder bouton detection and measurement. Therefore, it is essential to optimize the layout of manual traces, ensuring that they accurately follow axon centerlines in the underlying image.

In this study, we used the optimization algorithm described in (*Chothani et al., 2011*). In this version of the algorithm, positions of trace nodes $r_k = (x_k, y_k, z_k)^T$ are updated to optimize a fitness function, which includes intensity integrated along the trace and a regularizing constraint on the positions of neighboring trace nodes:

$$\mathcal{F}(\{r_k\}) = \sum_k \left( \frac{1}{\lambda} \sum_m I(l_m) \frac{e^{-\frac{\|r_k - l_m\|^2}{2R^2}}}{(2\pi)^{3/2} R^3} \left( 1 - \frac{2}{3} \frac{\|r_k\|^2}{R^2} \right) - \frac{\alpha\lambda}{2} \sum_{k'} \|r_k - r_{k'}\|^2 \right) \qquad (11)$$

Here, vectors $l_m$ denote the positions of voxel centers in the image stack, index $k'$ enumerates the neighbors of node $k$, parameter $\lambda$ denotes the average density of nodes in the trace (number of nodes per voxel), and Lagrange multiplier $\alpha > 0$ controls the stiffness of the trace. The first term in this expression represents convolution of the image with the Laplacian of Gaussian filter of size $R$, and due to the fast decay of the Gaussian factor, summation in this term can be restricted to a small number of voxels in the vicinity of the trace. The following parameter values were used to produce the results of this study. $R$ was set to three voxels (roughly equal to axon diameter in the images), $\lambda$ was set to 0.5, and $\alpha$ equaled 0.001. The fitness function was maximized with Newton's method as

previously described (*Gala et al., 2014*). Trace node positions were synchronously updated at every iteration step, and optimization terminated when the relative change in $\mathcal{F}$ fell below $10^{-6}$.

Results of this optimization procedure (*Figure 2—figure supplement 1B*) show marked improvement over manual traces. The optimized traces are smoother, closer to one another in $z$, and follow the underlying axon intensity in the image more accurately. Following optimization, all traces were subdivided to a higher density of nodes, $\lambda = 4$.

## Generation of axon intensity profiles

Parameters of filters used to generate the axon intensity profiles, *Equation (1)*, were chosen in the following way. The $xy$ size of the multi-scale $LoG_{xy}$ filter was chosen to span bouton sizes observed in 2PLSM images, $R_{xy} \in [1.5, 3.0]$ voxels (0.39 μm to 0.78 μm). Since the observed bouton size in the $z$ dimension is dominated by the point spread function of the microscope, a single size was chosen for the $z$ component of the filter, $R_z = 2$ voxels (1.6 μm). Upon convolution with the image, this multi-scale filter returns the maximum intensity calculated over the specified range of sizes. The Gaussian filter, $G$, was chosen to have a fixed size of $R = 2$ voxels in all three dimensions. This size was selected to be roughly equal to the typical axon shaft radius observed in the images (*Figure 2—figure supplement 1A*).

*Figure 2—figure supplement 1C* shows $LoG_{xy}$ profiles for the five manual traces from *Figure 2—figure supplement 1A*. Small differences in the layout of manual traces can lead to significant variability in intensity profiles, which is undesirable. Trace optimization reduces this variability to an extent undetectable by visual inspection, *Figure 2—figure supplement 1D*.

## Detection of putative boutons

Peak detection algorithm, *Equation (2)*, was initialized with a number of foreground peaks, $N_f = \lceil L/0.5\ \mu\mathrm{m} \rceil$, which is much larger than the expected number of putative boutons. In this expression, $\lceil\ \rceil$ denotes the ceiling function and $L$ is the trace length in micrometers. The number of background peaks, $N_b = \lceil L/25\ \mu\mathrm{m} \rceil$, was determined based on the observed spatial scale of background variability. The peaks were initially distributed uniformly along the entire length of the trace. Both, Gaussian and Lorentzian peak functions were tested, but only the former was used in this study as it provided a better fit to intensity profiles as judged by the value of the objective function.

The objective function was minimized with the gradient descent method. Gradient steps were taken simultaneously along the $\mu_j^f$, $\mu_k^b$, $a_j^f$, $a_k^b$, $\sigma_j^f$, and $\sigma_k^b$ dimensions. At every gradient step, parameters that moved outside the bounds specified in *Equation (2)* were set to these bounds. Upon convergence, that is, when the relative change in the value of the objective function became less than $10^{-6}$, one small foreground peak was eliminated or a pair of closely positioned foreground peaks was merged, and the resulting set of peaks was re-optimized. This procedure was continued until there were no peaks left that passed the following heuristic criteria: (1) a single foreground peak is marked for elimination if the peak amplitude is less than 0.3, and (2) a pair of overlapping foreground peaks is marked for merger if distance between the peaks is less than 1.0 μm or overlap area of the peaks exceeds 50% of either area. Here, the threshold of 0.3 was set to exclude small profile peaks that result from fluctuations of intensity along the axon. This threshold was chosen to be significantly lower than the mean profile intensity, which is one according to *Equation (6)*. Distance threshold of 1.0 μm (size of a small bouton, see e.g. *Figure 3*) and threshold on inter-peak overlap were introduced to merge peaks corresponding to the same varicosity.

## Matching putative boutons across imaging sessions

Custom software was used to match putative boutons over time. Because most large boutons remains stable over weeks to months, they can serve as fiducial points to match the remaining boutons. Here, piecewise linear registration of normalized intensity profiles was performed based on few fiducial boutons marked by users across imaging sessions (*Figure 6—figure supplement 1*). This was followed by matching the remaining putative boutons with a greedy, distance-based algorithm. All results were then validated with visual inspection. For putative boutons detected in some, but not all imaging sessions the missing bouton weights were filled in with the normalized intensity profile values at the corresponding registered positions. This procedure was performed for axons

traced by different users (*Figure 2—figure supplement 2*), axons repeatedly imaged under different conditions (*Figure 5*), and axons monitored in the long-term imaging experiment (*Figure 6*).

## Quantifying precision of bouton detection methodology

Small inaccuracies in layout of axon traces can introduce errors in bouton detection. Therefore, it is essential to verify that trace optimization can guarantee sufficiently high precision in bouton detection. For this test, putative boutons were detected automatically, both with and without trace optimization, on the same set of 16 axon segment traced manually by five users. In the absence of trace optimization, a total of 578 candidate bouton sites were identified and matched across all five user traces (*Figure 2—figure supplement 2A*). A candidate bouton site here is defined as a position on an axon where a putative bouton was detected based on at least one user trace. For the majority of candidate bouton sites, 348 (60%), a putative bouton was present in all five user traces, and thus, there was a full consensus in bouton detection. However, 141 (24%) candidate bouton sites had one conflict and 89 (16%) had two. A conflict is defined as a dissent from the majority, and given five traces, the maximum number of conflicts is two. In contrast, after trace optimization, 474 candidate bouton sites were detected, of which 409 (86%) had full consensus, 39 (8%) had one conflict, and 26 (6%) had two (*Figure 2—figure supplement 2B*). With trace optimization, it was possible to obtain complete agreement in all putative boutons of weights greater than 2.5. This is a marked improvement over bouton detection with no trace optimization where 17 putative boutons of weights greater than 2.5 had one or more conflicts.

Tracing inaccuracies also affect the weights of detected putative boutons. *Figure 2—figure supplement 2C*, shows that inaccurate manual traces can contribute to bias (slope difference from 1) and variance (deviation from the best fit line) in bouton weight, and that trace optimization significantly reduces such errors. To quantify the uncertainty in bouton weight we calculated root mean square (RMS) weight difference for putative boutons detected with no conflicts (*Figure 2—figure supplement 2D*). As was expected, trace optimization dramatically reduces this uncertainty in the entire range of bouton weights.

We would like to point out that some amount of uncertainty in bouton detection (*Figure 2—figure supplement 2B*) and measurement (*Figure 2—figure supplement 2D*) remains even after trace optimization. However, as shown in *Figure 5A–C* this uncertainty of the method is much smaller than variability originating from the experimental sources, and therefore, it is not a limiting factor in structural plasticity measurements.

## Implementation

Axons of fluorescently labeled neurons were traced by using the manual tracing module of NCTracer software (http://www.neurogeometry.org). Custom software written in MATLAB, BoutonAnalyzer (*Gala et al., 2017*; copy archived at https://github.com/elifesciences-publications/BoutonAnalyzer), was used to optimize the traces, generate intensity profiles, detect putative boutons, and match these boutons across multiple user traces and imaging sessions. BoutonAnalyzer enables the user to visually inspect the results of bouton detection and matching and edit them if necessary. The NeuroMorph Centerline Processing and NeuroMorph Measurement tools were used to measure axon cross-section areas, bouton and mitochondrial volumes, and PSD surface areas (*Jorstad and Knott, 2017*). A copy is archived at https://github.com/elifesciences-publications/NeuroMorph.

## Data availability

Data used in the CLEM experiment are available in the Dryad Digital Repository (https://dx.doi.org/10.5061/dryad.3q50t). This dataset includes a 2PLSM image stack in TIFF format (*Figure 3A*), optimized traces of four axon segments in SWC format (*Figure 3A*), and 3D EM reconstructions of these axons in Blender format (*Figure 3C*).

## Acknowledgements

We are grateful to Jerome Blanc, Raeann Dalton, Rammy Dang, and Chi Zhang for their help in tracing neurons and annotating boutons. This work was supported by the NIH grant R01 NS091421 and AFOSR grant FA9550-15-1-0398 to AS, Swiss National Science Foundation (SNF) grants 331003A_153448 and 51NF40_158776 (National Centre for Competence in Research, SYNAPSY),

and the International Foundation for Research on Paraplegia (chair Alain Rossier) to AH, and SNF grant CRSII3-154453 to AH and GK.

## Additional information

### Funding

| Funder | Grant reference number | Author |
| --- | --- | --- |
| National Institutes of Health | R01 NS091421 | Armen Stepanyants |
| Air Force Office of Scientific Research | FA9550-15-1-0398 | Armen Stepanyants |
| Schweizerischer Nationalfonds zur Förderung der Wissenschaftlichen Forschung | 331003A_153448 | Anthony Holtmaat |
| Schweizerischer Nationalfonds zur Förderung der Wissenschaftlichen Forschung | CRSII3-154453 | Graham Knott Anthony Holtmaat |
| Schweizerischer Nationalfonds zur Förderung der Wissenschaftlichen Forschung | 51NF40_158776 | Anthony Holtmaat |
| International Foundation for Research in Paraplegia | Chair Alain Rossier | Anthony Holtmaat |

The funders had no role in study design, data collection and interpretation, or the decision to submit the work for publication.

### Author contributions

Rohan Gala, Conceptualization, Software, Formal analysis, Validation, Methodology, Writing—original draft, Writing—review and editing; Daniel Lebrecht, Conceptualization, Resources, Validation, Investigation, Methodology, Writing—review and editing; Daniela A Sahlender, Resources, Validation, Investigation, Writing—review and editing; Anne Jorstad, Resources, Software, Validation, Investigation, Writing—review and editing; Graham Knott, Resources, Investigation, Software, Supervision, Funding acquisition, Validation, Methodology, Project administration, Writing—review and editing; Anthony Holtmaat, Conceptualization, Resources, Supervision, Funding acquisition, Validation, Methodology, Writing—original draft, Project administration, Writing—review and editing; Armen Stepanyants, Conceptualization, Formal analysis, Supervision, Funding acquisition, Validation, Methodology, Writing—original draft, Project administration, Writing—review and editing

### Author ORCIDs

Rohan Gala  http://orcid.org/0000-0003-1872-0957
Daniela A Sahlender  http://orcid.org/0000-0002-0510-9529
Anne Jorstad  http://orcid.org/0000-0002-6438-1979
Graham Knott  https://orcid.org/0000-0002-2956-9052
Anthony Holtmaat  http://orcid.org/0000-0002-7577-0769
Armen Stepanyants  http://orcid.org/0000-0002-9387-2320

### Ethics

Animal experimentation: All experiments were performed according to the guidelines of the Swiss Federal Act on Animal Protection and Swiss Animal Protection Ordinance. The ethics committee of the University of Geneva and the Cantonal Veterinary Office of Geneva, Switzerland (approval code GE/61/17) approved all experiments.

### Decision letter and Author response

Decision letter https://doi.org/10.7554/eLife.29315.017
Author response https://doi.org/10.7554/eLife.29315.018

# Additional files

## Supplementary files
• Transparent reporting form
DOI: https://doi.org/10.7554/eLife.29315.014

## Major datasets
The following dataset was generated:

| Author(s) | Year | Dataset title | Dataset URL | Database, license, and accessibility information |
|---|---|---|---|---|
| Gala R, Lebrecht D, Sahlender DA, Jorstad A, Knott G, Holtmaat A, Stepanyants A | 2017 | Data from: Computer assisted detection of axonal bouton structural plasticity in in vivo time-lapse images | http://dx.doi.org/10.5061/dryad.3q50t | Available at Dryad Digital Repository under a CC0 Public Domain Dedication |

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
