## [Decision Letter]

Thank you for submitting your article "Computer assisted detection of axonal bouton structural plasticity in in vivo time-lapse images" for consideration by *eLife*. Your article has been reviewed by three peer reviewers, and the evaluation has been overseen by Eve Marder as the Senior Editor and Reviewing Editor. The following individuals involved in review of your submission have agreed to reveal their identity: Sen Song (Reviewer #1); Albert Cardona (Reviewer #2). A further reviewer remains anonymous.

The reviewers have discussed the reviews with one another and the Reviewing Editor has drafted this decision to help you prepare a revised submission.

Summary:

The authors proposed new algorithms to analyze in vivo two-photon time-lapse images for axonal bouton structural plasticity and validated the method against EM reconstructions of the same tissue. The manuscript reports on a new method for quantifying synaptic boutons in light-microscopy images of fluorescently labeled axons, and validates the method and parameter choices by using correlative light-electron microscopy, providing the nanometer-scale reconstruction of axons and synaptic surfaces of the micrometer-scale light-microscopy images. The method models sources of variability, both controllable and non-controllable, while also acknowledging and including the inherent ambiguity of synaptic boutons in predicting the presence of a synaptic surface (not all boutons house synapses or, as the author's put it, postsynaptic densities). The normalization approach is appropriate and accurate, and its structure reveals the careful, neuroanatomy-based reasoning that led to its design. The method applies to both small varicosities and varying imaging conditions, including low-intensity and low-contrast conditions, and reports a precision and recall good enough that, together with the modeling of variability and the ability to track varicosities over time on the same axons, essentially sets the stage for fully-automated estimation of synaptic weights from light-microscopy images of brain tissue for chronic studies.

Essential revisions:

1) One of the issues that arose during the review is that some of your previously published methods are inaccessible without paying exorbitant fees to other publishers. Consequently, please include all of the methods you used in detail here, so that this paper stands alone and exists in the OPEN ACCESS world with all of its methods. *eLife* does not restrict the space that can be used for methods, so it will be a service to the community to have the methods here. Among specific issues raised by the reviewers:

a) There is an insufficiently detailed EM fixation and counterstaining protocol: no concentrations given for aldehydes, osmium, uranyl acetate, and no specifications on the individual steps.

b) More detail on the trace optimization method.

c) No open source code repository is reported, which would be critical to distributing the software to facilitate the application of this novel method and provide commit hashes for reference when reporting the use of the method, towards ensuring reproducibility of results. Attaching the source code to the paper is an old practice from when public source code repositories were rare or hard to setup, and while the practice ensures archiving of the source code, it is not conducive to further corrections, extensions or modifications. Which source version control repository hosts the code, and what hash / commit number corresponds to the attached code zip file? This is critical for *eLife* publication!

2) Please address the issue of whether the laplacian of gaussian filter for detection of boutons introduces biases in estimating bouton intensities. Figure 4 addresses some of those issues, but the profile intensities there are after several further steps of processing. This issue need to be explored in a bit more depth by perhaps comparing with some other methods. Address how the parameters were chosen for the LOG filter.

3) One reviewer says, "The detection of putative boutons based on the Backward-Stepwise Subset Selection method is also novel. I think the fitting of foreground peaks using a variable size gaussian filter is a very good idea. But I am less convinced of fitting the background peaks also with Gaussian functions. There is no a priori reason why the background is also peaky. Have the authors explored other functions for fitting the background. What is the physical intuition of the background peaks?"

4) For the shaft intensity, why did you choose the positions where the boutons were present for measurement? It seems more intuitive to measure from areas which are devoid of boutons. The current method seems more like an average bouton intensity. Normalizing the image by this kind of quantity suffers from one drawback, which is if a large fraction of boutons are gained or lost, this might lead to a bias.

5) Fitting the bouton gain and loss functions using a probabilistic function is a very good idea. But it is not very clear how this could benefit final biological measurements and interpretation. Is it useful for taking/getting more meaningful interpretation for different amounts of measurement noises? Some discussion of this would be useful.

6) The correlation with EM looks quite good, but is it better than previous methods? Some comparison would be useful.

Optional: One reviewer felt that one potential way to increase the impact is to demonstrate its robustness for data previously published by other labs. Other reviewers did not feel this was necessary, but we wanted you to think about this.

---

## [Author Response]

Essential revisions:1) One of the issues that arose during the review is that some of your previously published methods are inaccessible without paying exorbitant fees to other publishers. Consequently, please include all of the methods you used in detail here, so that this paper stands alone and exists in the OPEN ACCESS world with all of its methods. eLife does not restrict the space that can be used for methods, so it will be a service to the community to have the methods here. Among specific issues raised by the reviewers:a) There is an insufficiently detailed EM fixation and counterstaining protocol: no concentrations given for aldehydes, osmium, uranyl acetate, and no specifications on the individual steps.b) More detail on the trace optimization method.c) No open source code repository is reported, which would be critical to distributing the software to facilitate the application of this novel method and provide commit hashes for reference when reporting the use of the method, towards ensuring reproducibility of results. Attaching the source code to the paper is an old practice from when public source code repositories were rare or hard to setup, and while the practice ensures archiving of the source code, it is not conducive to further corrections, extensions or modifications. Which source version control repository hosts the code, and what hash / commit number corresponds to the attached code zip file? This is critical for eLife publication!

We have expanded the Materials and methods section to include additional details related to EM methods and trace optimization. The source code of BoutonAnalyzer is now hosted on GitHub, and the version tagged v1.0 (commit number 531a792) is used in conjunction with the manuscript.

2) Please address the issue of whether the laplacian of gaussian filter for detection of boutons introduces biases in estimating bouton intensities. Figure 4 addresses some of those issues, but the profile intensities there are after several further steps of processing. This issue need to be explored in a bit more depth by perhaps comparing with some other methods. Address how the parameters were chosen for the LOG filter.

Axon intensity profiles are produced in two steps, (1) sliding the LoG_xy_ filter along the trace and (2) normalizing the result with the shaft intensity. In this heuristic procedure, normalization was included to remove bias due to axon specific calibers and fluorophore expression levels. CLEM analysis shows that bouton weights resulting from this procedure are well correlated with bouton volumes, and therefore, we have no reason to think that the LoG_xy_ filter biases the results. It is difficult to be more conclusive here, because the size of the CLEM dataset is relatively small.

Nevertheless, we explored this issue in some detail and now provide comparisons with other plausible bouton detection strategies in our manuscript. We added Figure 4—figure supplement 1 showing intensity profiles obtained without filtering (raw voxel intensities along the trace), and produced with mean, Gaussian, and median filters of different but fixed sizes. This figure shows that in the absence of filtering, profiles appear jagged, which hinders bouton detection. At the other extreme, large filters suppress small boutons and merge closely positioned boutons. By tuning sizes of the considered filters we were able to obtain profiles that appear similar to those of multi-scale LoG_xy_, although with higher baselines. Still, we prefer LoG_xy_ because its negative surround produces near-zero baselines, which simplifies bouton detection. In addition, the multi-scale feature of this filter eliminates the need for size tuning.

We also examined the combined effect of filtering and normalization on bouton measurement by comparing the amplitudes of profile peaks to corresponding bouton volumes. Figure 4—figure supplement 1 shows that the overall procedure described in the manuscript leads to the highest correlation as compared to the best of the considered alternatives.

We would like to emphasize that we did not attempt to find the best procedure to fit the CLEM dataset. Some other combination of filter type, normalization, and parameters may perform marginally better, but this would likely be a case of over-fitting the data. Instead, our procedure was designed based on theoretical considerations, and the parameters were chosen based on observed bouton sizes. These points are now mentioned in the last paragraph of the Results subsection “Validation of LM-based bouton detection methodology with EM”.

3) One reviewer says, "The detection of putative boutons based on the Backward-Stepwise Subset Selection method is also novel. I think the fitting of foreground peaks using a variable size gaussian filter is a very good idea. But I am less convinced of fitting the background peaks also with Gaussian functions. There is no a priori reason why the background is also peaky. Have the authors explored other functions for fitting the background. What is the physical intuition of the background peaks?"

Axon background intensity defined in the manuscript is the intensity an axon intensity profile would have in the absence of boutons. If bouton density were sufficiently low, background intensity could be deduced from inter-bouton portions on the axon. However, for high bouton densities, which is often the case for cortical axons, background intensity has to be disentangled from bouton contributions. We did this with a two-step fitting procedure described in the manuscript. To fit the background, we used multiple, relatively wide Gaussian peaks (σ_k_^b^ ≥ 20 μm, [Disp-formula equ2]). This was done to account for the possibility of variations in background intensity over relatively large distances along the axon (>> 1 μm or bouton size). Such variations in intensity may result from changes in axon caliber or changes in density of fluorescent molecules. Gaussian peaks were used for convenience, but any other slowly varying function would work as well. We would like to emphasize that the resulting background, which is a sum of multiple overlapping Gaussians, varies very smoothly and excludes intensity of putative boutons as shown in Figure 2 (red line). We now emphasize these points in the subsection “LM-based bouton detection and measurement of structural changes”.

4) For the shaft intensity, why did you choose the positions where the boutons were present for measurement? It seems more intuitive to measure from areas which are devoid of boutons. The current method seems more like an average bouton intensity. Normalizing the image by this kind of quantity suffers from one drawback, which is if a large fraction of boutons are gained or lost, this might lead to a bias.

Our method was designed to produce robust results, regardless of bouton density. In particular, when density of boutons is high, e.g. Figure 2, identifying areas devoid of boutons can be next to impossible. Therefore, we infer shaft intensity by using a fitting procedure which disentangles it from bouton contributions. Specifically, axon intensity profiles are fitted simultaneously with foreground and background peak functions, and only the background peak functions are used to determine the shaft intensity (red line in Figure 2). The resulting shaft intensity is independent of bouton density, and, thus, it is expected to be insensitive to changes in density arising from bouton addition or elimination. We now clarify this point in the subsection “LM-based bouton detection and measurement of structural changes”.

5) Fitting the bouton gain and loss functions using a probabilistic function is a very good idea. But it is not very clear how this could benefit final biological measurements and interpretation. Is it useful for taking/getting more meaningful interpretation for different amounts of measurement noises? Some discussion of this would be useful.

When the measured bouton weight is well above/below w_threshold_, it may be said with confidence that an LM bouton is present/absent. In such cases, deterministic definition of a bouton is sufficient, and it is in agreement with [Disp-formula equ10]. In practice, because the distribution of bouton weights is unimodal (see Figure 2—figure supplement 2), a large fraction of weights will lie within the range of uncertainty of w_threshold_, regardless of its value. Such ambiguous boutons could be discarded, which may bias biological interpretation. Alternatively, they could be treated probabilistically as described in the manuscript, which is our preferred method. We think that this method may offer distinct benefits for biological interpretation. One could consider the following specific examples of biological measurements where a probabilistic approach is essential. (1) Estimate of expected number of LM boutons. Here, since bouton weight distribution is non-uniform (monotonically decreasing), one cannot simply count the number of weights above w_threshold_ to estimate the expected number of LM boutons, even for a large sample size. This count would result in an underestimate, as many more measured sub-threshold weights may have a true weight > w_threshold_ than the other way around. Thus, it is essential to add LM bouton probabilities defined by [Disp-formula equ10]. (2) The same argument holds for calculations of the expected numbers of added, eliminated, potentiated, and depressed boutons. (3) Most bouton structures are stable and do not change substantially over days-to-weeks. Therefore, changes in weight often lie within the range of measurement uncertainty and can only be described probabilistically according to [Disp-formula equ11]. More importantly, probabilistic description is required for calculation of error-bars and establishing significance of experimental measurements. This is absolutely essential for correct biological interpretation of results. These points are now mentioned in the last paragraph of the Discussion.

6) The correlation with EM looks quite good, but is it better than previous methods? Some comparison would be useful.

We are aware of only one publicly available software tool for bouton detection and measurement, EPBscore (Song et al., 2016), which we cite in our manuscript. This software generates an axon intensity profile based on a combination of fixed-size Gaussian and median filters, and subsequently normalizes it by its median. It is designed to work well on isolated axons. However, when multiple axons in close proximity cross paths, which is the case in our CLEM dataset, EPBscore does not provide a utility to segment individual axons or to obtain accurate traces of axons in 3D. Thus, unfortunately, we were not able to successfully use EPBscore here due to a relatively high density of labeled axons in our dataset (see e.g. Figure 1).

There are several other publications related to bouton detection and measurement. Some of these studies use local shaft intensity for normalization, but this can only be done for low bouton density axons. Other studies use complex sequences of thresholds to determine normalization, which is unlikely to generalize on different types of images. There are also studies that normalize by the mean or median axon intensity. These normalizations are reasonable for low bouton density axons, but can otherwise bias the results. Absence of publicly available tools and/or source codes precluded direct comparisons of these studies with our method.

Instead, we incorporated several of the above strategies into our bouton detection framework and now directly compare them to the method described in this study (see Figure 4—figure supplement 1 and related discussion in response to Essential revision 2).

Optional: One reviewer felt that one potential way to increase the impact is to demonstrate its robustness for data previously published by other labs. Other reviewers did not feel this was necessary, but we wanted you to think about this.

We would like to test the software on other CLEM datasets, but we are not aware of any datasets that are publicly available. Based on our knowledge of the literature, such data are extremely rare and usually include small numbers of reconstructed en passant boutons. For example, one of the largest CLEM datasets we know (Grillo et al., PNAS 2013), contains only 9 en passant boutons. If the reviewers know of available CLEM data, we would be happy to perform the analyses and include the results in the manuscript.